# ✻ DataDecide

## How to Predict Best Pretraining Data with Small Experiments

**Ian Magnusson** [* 1 2]  **Nguyen Tai** [* 3]  **Ben Bogin** [* 1]  **David Heineman** [1]  **Jena Hwang** [1]  **Luca Soldaini** [1]  **Akshita Bhagia** [1]
**Jiacheng Liu** [1 2]  **Dirk Groeneveld** [1]  **Oyvind Tafjord** [1]  **Noah A. Smith** [1 2]  **Pang Wei Koh** [1 2]  **Jesse Dodge** [1]

## Abstract

Because large language models are expensive to pretrain on different datasets, using smaller-scale experiments to decide on data is crucial for reducing costs. Which benchmarks and methods of making decisions from observed performance at small scale most accurately predict the datasets that yield the best large models? To empower open exploration of this question, we release models, data, and evaluations in DATADECIDE—the most extensive open suite of models over differences in data and scale. We conduct controlled pretraining experiments across 25 corpora with differing sources, deduplication, and filtering up to 100B tokens, model sizes up to 1B parameters, and 3 random seeds. We find that the ranking of models at a single, small size (e.g., 150M parameters) is a strong baseline for predicting best models at our larger target scale (1B) ($\sim 80\%$ of comparisons correct). No scaling law methods among 8 baselines exceed the compute-decision frontier of single-scale predictions, but DATADECIDE can measure improvement in future scaling laws. We also identify that using continuous likelihood metrics as proxies in small experiments makes benchmarks including MMLU, ARC, HellaSwag, MBPP, and HumanEval $> 80\%$ predictable at the target 1B scale with just 0.01% of the compute.

## 1. Introduction

The cost of training large language models (LMs) necessitates methods of trying out options at small scale, but it also makes it expensive to validate the accuracy of development decisions made with such methods. We focus on the question of choosing between pretraining datasets to use—one of the most impactful development decisions. Common practice (e.g., Li et al., 2024) uses a single, small scale of experiments to cheaply test pretraining data intended for larger-scale models, where scale is determined by number of model parameters and training tokens. The other predominant approach is to fit scaling laws (Kaplan et al., 2020; Hoffmann et al., 2022; Choshen et al., 2024) to the trend in performance observed over multiple small scales, with recent work extending this to the prediction of downstream performance instead of language modeling loss (Gadre et al., 2024; Dubey et al., 2024; Bhagia et al., 2024).

So far decision-making approaches have only been validated without observing the counterfactual outcome, either by producing a single large model on the chosen decision with impressive performance or by low error in predicting the magnitude of observed performance of a small number of large models. Knowing what amount of error in predicting performance over scale is a low enough to actually make a correct decision among datasets, requires a suite of comparable models trained on many datasets. Although a wide variety of open-source pretraining corpora are available, the scaling behavior of data is difficult to assess from off-the-shelf models that vary simultaneously in data, optimizer, and modeling decisions.

To make it possible to empirically study what methods make the best decisions over data, we build DATADECIDE[1]—a suite of models we pretrain on 25 corpora up to 100B tokens, over 14 different model sizes ranging from 4M parameters up to 1B parameters (more than 30K model checkpoints in total). We evaluate all models across a suite of 10 downstream tasks and calculate how accurately small models predict which pretraining corpora lead to better performance

---
[*]Equal contribution  [1]Allen Institute for AI  [2]Paul G. Allen School of Computer Science & Engineering, University of Washington  [3]University of Pennsylvania. Correspondence to: Ian Magnusson <ianmag@cs.washington.edu>.

*Proceedings of the $42^{nd}$ International Conference on Machine Learning*, Vancouver, Canada. PMLR 267, 2025. Copyright 2025 by the author(s).

---
[1]DataDecide collection on HuggingFace

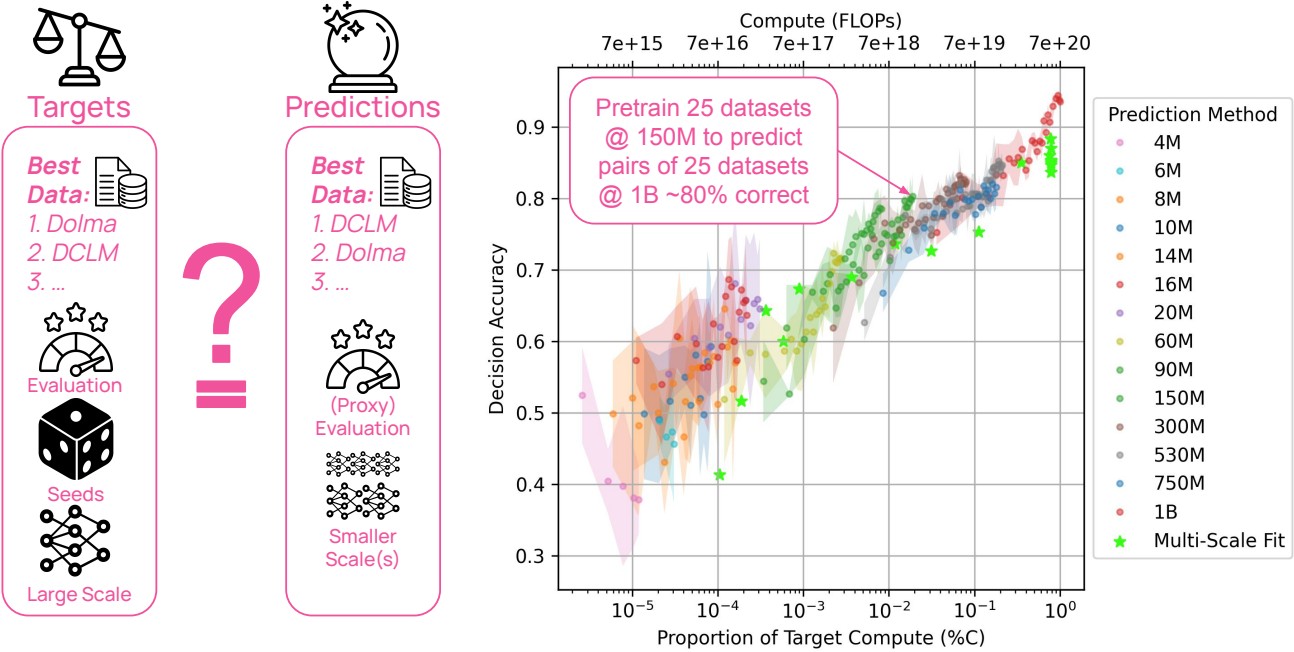

Figure 1. Which pretraining data to use? Ideally, compare performance of large models with fixed configurations averaged over random seeds (left). In practice, cheaper, smaller-scale experiments are used (center). Here DATADECIDE measures accuracy of pairwise decisions between 25 pretraining corpora to find efficient prediction methods (right).

at our largest scale. Our conclusions provide practical recommendations for the best benchmarks, prediction methods, and metrics to use to make decisions.

We call the 25 corpora we train on *data recipes* as they range across popular corpora including Dolma (Soldaini et al., 2024), DCLM (Li et al., 2024), RefinedWeb (Penedo et al., 2023), C4 (Raffel et al., 2019), and FineWeb (Penedo et al., 2024) as well as combinations of interventions on these datasets such as source mixing, deduplication, and filtering. Previous work has considered only 2 (Biderman et al., 2023) or 6 recipes (Magnusson et al., 2024; Brandfonbrener et al., 2024). We also offer a novel affordance by including 3 random seed reruns for even our largest runs, to help quantify whether variation occurs due to random initialization and data order or differences in the distribution of data.

Concretely, DATADECIDE allows analyses such as Figure 1 (right), which shows the relationship between compute used to predict a ranking of datasets and how accurately that ranking reflects mean performance over 3 seed runs (quantified here by OLMES; Gu et al., 2024) for models fully trained on those datasets at the target (1B) scale. We measure the accuracy of decisions as the percent of compared pairs of datasets where the prediction identifies the correct winner. Each point represents the average decision accuracy of a given method over 3 prediction attempts using small models with different random seeds, and shading shows standard

deviation.

Measuring the tradeoff of compute cost to better decisions lets us make the following recommendations about small experiments for making data decisions:

- §3.1 – The amount of compute you need to allocate for a given decision accuracy depends heavily on task. MMLU and ARC are much cheaper to predict than HellaSwag and some tasks such as SocialIQA are difficult to predict at all scales.

- §3.2 – 8 baseline scaling law methods do not exceed the compute to decision accuracy frontier set by ranking single scale experiments.

- §3.3 – At small scales, continuous metrics using answer likelihood are better or equivalent predictors of decisions than using the same discrete accuracy target metric.

- §3.4 – Better decisions can be explained in part by low run-to-run variance and a wide spread of benchmark performance values for different data, traits which can be improved by proxy metrics.

Future research can extend DATADECIDE with little extra compute by running new evaluations on our checkpoints, pretraining additional small models to compare against the large target models we provide, or trying new prediction

| Source / Recipe | Description |
| --- | --- |
| **Dolma1.7** *Original, No code, No math/code, No Reddit, No Flan* | A 2.3T-token corpus (Dolma 1.7 Soldaini et al., 2024) sampling common LM sources for open research. We ablate code, math/code, Reddit, or Flan subsets. |
| **Dolma1.6++** *Original* | Dolma 1.6 plus additional sources from Dolma 1.7: RedPajama's arxiv subset, openwebmath, algebraic stack, flan, starcoder, falcon. |
| **C4** *Original* | The C4 dataset (Raffel et al., 2019) as prepared in Dolma 1.7, heuristically filtered from the April 2019 Common Crawl. |
| **FineWeb-Pro** *Original* | The FineWeb Pro corpus (Zhou et al., 2024), featuring model-driven data cleaning on FineWeb. |
| **FineWeb-Edu** *Original* | The deduplicated FineWeb-Edu subset of SmolLM-Corpus (Ben Allal et al., 2024), focused on educational web pages. |
| **Falcon** *Original* | The Falcon RefinedWeb corpus (Penedo et al., 2023) in Dolma 1.7, derived from Common Crawl through June 2023 and more aggressively filtered/deduplicated than C4. |
| **Falcon+CC** *Original, QC 10%, QC 20%, QC Orig 10%, QC Tulu 10%* | Falcon and Dolma 1.7's Common Crawl. We quality filter to top 10% or 20% documents with reproduced or original (Li et al., 2024) filter or retrain filter on pre-release version of Tulu-v3 (Lambert et al., 2024). |
| **DCLM-Baseline** *Original, QC 7% FW2, QC 7% FW3, QC FW 3%, QC FW 10%, QC 10%, QC 20%* | A SOTA Common Crawl corpus using best ablated deduplication, cleaning heuristics, and quality filter. We quality filter to top 7% of DCLM classified documents and further take 2+ or 3+ scores with FineWeb-edu classifier; or filter to top 3% or 10% with FineWeb-edu classifier; or take top 10% or 20% with reproduced DCLM classifier. |
| $\lambda\%$ **DCLM-Baseline** $+ 1 - \lambda\%$ **Dolma1.7** | Fractional combinations of Dolma1.7 and DCLM-Baseline mixing different proportions of the two datasets for $\lambda \in \{25\%, 50\%, 75\%\}$. |

*Table 1.* DATADECIDE enables the study of data differences over scales through controlled pretraining experiments on 25 data recipes. These take different source datasets and apply interventions from ablating domains, deduplication, mixing, to quality filtering with different classifiers and thresholds. We release all pretraining corpora, as well as models trained on each recipe and each of the 14 model configurations in Table 2 with 3 random seeds.

methods with lightweight manipulations such as smoothing and curve fitting on top of our released evaluation results.

## 2. Methods

Our aim is to empirically test the predictability of downstream performance at a larger, target scale using small experiments. We describe DATADECIDE §2.1, the prediction methods we examine §2.2, the metrics we use to assess predictions §2.3, how we measure downstream performance §2.4, and proxy metrics for our performance evaluations §2.5. We will release all models, checkpoints, pretraining corpora, and evaluations.

### 2.1. The DATADECIDE Suite

We pretrain a suite of 1,050 models using 25 data recipes $\times$ 14 model scales $\times$ 3 random seeds for initialization and data order. Table 1 describes the 25 data recipes included in DATADECIDE that aim to provide coverage of common data preparation choices such as deduplication, ablating domains, mixes of existing datasets, as well as quality filters with different implementations, training data, and thresholds for quality classifiers.

We select a token to parameter ratio of 100, which at $5\times$

"Chinchilla" ($5 \times C$) optimal ratio (Hoffmann et al., 2022) captures the typical overtraining favored for inference savings.

All 1B (target size) models have 3 full reruns with different seeds, while other model sizes have second and third seed runs that are terminated early after $25\%$ of the target compute budget. We train the 1B reruns all the way to completion to allow our target "gold" predictions to account for run-to-run variance in evaluations due to weight initialization and data order. For instance, we find that the standard deviation between runs at the 1B $5\times C$ scale can be as high as $2\%$ points of accuracy for some recipes on most tasks. Meanwhile, at the non-target scales we wish to make predictions with a small fraction of the target compute, so we avoid reruns that would use an impractically large prediction budget.

Whether for extrapolating scaling laws or ranking single scale experiments, it is important to select reasonable hyperparameters for each scale to avoid confounding in performance differences that are simply due to suboptimal hyperparameters. We use OLMo's *model ladder* (Groeneveld et al., 2024; OLMo et al., 2025; Bhagia et al., 2024) to programmatically create LM pretraining configurations for a specified parameter size and token-parameter ratio to enable

running a grid of model scaling experiments. The model ladder uses heuristics from the literature (Porian et al., 2024) to set global batch size and learning rate based on scaling factors. The hyperparameters that determine parameter count (layers, hidden dimension, number of heads, MLP dimension) were handpicked by OLMo developers for each scale to achieve the desired number of parameters. Appendix Table 2 details the configurations of all our models.

## 2.2. Prediction Methods

Broadly, there are two approaches in the literature to predicting large-scale performance based on small-scale experiments. We use straightforward implementations of each to assess where they succeed and fail at making decisions about which data recipes to use.

**Ranking Single Scale Experiments (Single Scale)** This simple approach is employed by work such as Li et al. (2024) and consists of running a set of ablations or experiments over data recipe options while holding constant all other modeling variables including scale. The winning data recipe by downstream accuracy (or proxies) at the small experimental scale is assumed to extrapolate to the target scale.

**Extrapolating Scaling Laws (Multi Scale)** Another approach to making decisions with predictions across scales used in works such as Dubey et al. (2024) is to fit scaling laws to *multiple* small experiments across a range of scales for each of the data recipes. The winning recipe is decided as the one whose scaling law shows the highest *extrapolated* performance at the target scale. Although scaling laws were first observed for language modeling loss (Kaplan et al., 2020; Hoffmann et al., 2022), they have been extended to predict downstream performance through a two-step approach that also fits a function from loss to downstream performance (Gadre et al., 2024; Bhagia et al., 2024). We follow a method from Bhagia et al. (2024). Their proposed approach incorporates separate parameters for number of model parameters and number of tokens trained to account for over or undertrained models. But as our suite only includes one token-parameter ratio, we use the simplified 3 parameter baseline, $L(C)$, as a first step which we chain with second step, $Acc(L)$, defined as follows where $A$, $\alpha$, $E$, $a$, $b$, $k$, $L_0$ are optimized parameters:

$$L(C) = \frac{A}{C^\alpha} + E \qquad (1)$$

$$Acc(L) = \frac{a}{1 + e^{-k(L-L_0)}} + b \qquad (2)$$

Following Bhagia et al. (2024) we fit Equation 1 only on observations of final, fully trained checkpoints as accounting for the learning rate schedule's impact on intermediate checkpoints would require further parameters in the equation increasing the required number of observations and

cost. To account for step-to-step noise in evaluation we average the last $10\%$ of checkpoints as the final observed loss. Equation 2, however, is fit on all observations including intermediate checkpoints. We explore variations for a total of 8 multi scale approaches defined in Appendix C; none of these make for substantially better decisions than the method defined in this section.

## 2.3. Prediction Metrics

Our predictive task is to forecast which of a pair of data recipes will perform better at some target scale based on small-scale experiments. We use the following metrics to measure the quality of these predictions.

**Prediction Error** Scaling laws literature (Bhagia et al., 2024; Gadre et al., 2024) typically evaluates success from predicted and actual downstream performance, using relative error ($\frac{|\text{predicted}-\text{actual}|}{\text{actual}} \times 100\%$) or absolute error ($|\text{predicted} - \text{actual}| \times 100\%$). We call these absolute or relative "prediction error" to distinguish from the following metric.

**Decision Accuracy** Unlike previous work, we also measure the impact of predictions on *decisions* about which data recipe is better than another. The metric we use to capture this is decision accuracy, an accuracy over all pairs of data recipes $A$ and $B$ where either $A$ or $B$ is defined as the correct winner based on which achieves higher performance at the target scale. This is nearly equivalent to Kendall's $\tau$, but ranges from 0 to 1. We define the target-scale winner based on mean downstream performance over 3 random seeds. Thus decision accuracy can be formalized as follows. Let $\mathcal{P}$ be the set of all data recipe pairs $(A, B)$ with observed mean performance $y_A, y_B$ and predicted performance $\hat{y}_A, \hat{y}_B$, respectively, then decision accuracy is:

$$\frac{1}{|\mathcal{P}|} \sum_{(A,B)\in\mathcal{P}} \mathbb{I}\big(\text{sign}(\hat{y}_A - \hat{y}_B) = \text{sign}(y_A - y_B)\big) \quad (3)$$

**Percent of Target Compute Budget** ($\%C$) We measure compute in terms of theoretical FLOPs following the simplifying assumption made in most scaling literature that the costs associated with training a model are captured well enough by FLOPs $= 6ND$, based solely on the number of parameters ($N$) and tokens trained ($D$) (Kaplan et al., 2020). We consider the efficiency of a prediction based on the ratio of the experimental budget and the target budget in FLOPs, $\%C = \frac{c}{C} \times 100\%$.

## 2.4. Performance Evaluation with OLMES

We use the OLMES suite of 10 multiple choice question answering benchmarks (Gu et al., 2024): MMLU (Hendrycks et al., 2021), HellaSwag (Zellers et al., 2019), ARC Challenge (Clark et al., 2018), ARC Easy (Clark et al., 2018),

PIQA (Bisk et al., 2020), CommonsenseQA (Talmor et al., 2019),SocialIQA (Sap et al., 2019), OpenBookQA (Mihaylov et al., 2018), BoolQ (Clark et al., 2019), and Wino-Grande (Sakaguchi et al., 2020). These tasks are well suited for the model scales we examine with all but BoolQ receiving non-trivial performance. Unless otherwise noted, we consider the macro average of these ten tasks. The underlying metric for each task is accuracy, for which OLMES specifies a different length normalization scheme per task. Our target "gold" rankings which we aim to predict are always based on the "cloze" formulation (CF) accuracy with curated normalization per task, which we refer to as ACCURACY. We diverge from OLMES only in that we make use of all available items in the specified split of each benchmark rather than subsampling them, to reduce variance over the task distribution.

Note that while we focus just on OLMES multiple choice evaluations in this work, our method of validating decisions made through predictions can be applied to other benchmarks. We chose these tasks based on their appropriateness to our range of model scales, and one would have to select different tasks when targeting a larger scale. Moreover, DATADECIDE could be used to identify new evaluations that are sensitive within our range of scales.

## 2.5. Proxy Metrics for Performance Evaluation

Previous work has noted how discrete metrics such as accuracy can cause jumps in performance across scale that otherwise see more predictable improvements with scale for continuous metrics (Schaeffer et al., 2023). We experiment with using continuous metrics at small scale as proxies of the accuracies selected by OLMES for each task (ACCURACY) at the target scale to improve decision accuracy. We use the following metrics: CORRECT PROB is the average probabilities of the correct continuations. MARGIN is the average difference between the probability of the correct continuation and the most likely incorrect continuation. NORM CORRECT PROB is the average probability of the correct continuation conditioned on the response being in the set of correct or incorrect continuations. TOTAL PROB is the average of the sum of probabilities of all correct and incorrect continuations. ACCURACY is the fraction of instances where the correct continuation has the highest probability. Each of these can be computed with likelihoods normalized by number of tokens or characters; unless otherwise specified we use character length normalization. Appendix Table 3 shows formal definitions.

## 3. Results

### 3.1. What is the best way to spend compute for data decisions?

*More compute makes better decisions. Decisions from intermediate checkpoints are as good as compute equivalent final checkpoints. The amount of compute needed to make good predictions varies between tasks. ARC and MMLU are predictable with much less compute than HellaSwag. The rest of OLMES tasks give markedly less reliable predictions across the scales we examine.*

First looking at the aggregation of all 10 OLMES tasks (Figure 1 right), we see that there is a positive and roughly log-linear relationship between experimental compute and decision accuracy. Specifically, this figure illustrates the relationship between the compute used for predicting best data recipes and the decision accuracy those predictions achieve against targets ranked by OLMES performance at the 1B scale. Each point represents the average decision accuracy over three runs with different random seeds, with shading indicating standard deviation. Points with the same color show all intermediate checkpoints from a given parameter size. The color shows each model size for predicting using ranking single scale experiments. The stars show predictions from extrapolating scaling laws using our default 3-parameter approach, the details of which are discussed further in §3.2.

The ease of prediction is greatly influenced by which evaluation benchmark we use. In Figure 2, we show the relationship of compute and decision accuracy for each of the tasks in OLMES individually. The predictive sensitivity of tasks at a given compute varies significantly, with ARC Easy being consistently predictable with 5 orders of magnitude less compute and BoolQ only reaching beyond trivial decision accuracy for intermediate checkpoints of the target runs. HellaSwag, SocialIQA, WinoGrande show distinct periods of insensitivity followed by roughly log-linear increase after hitting some compute threshold.

### 3.2. How does extrapolating scaling laws compare to ranking single scale experiments?

*A selection of 8 baseline scaling law methods are no more efficient than ranking single scale experiments. Future scaling law methods can be assessed on DATADECIDE.*

Figure 3 contrasts different approaches to fitting scaling laws over multiple scales of small experiments. Each of the 8 approaches is shown in a different color. Multi-scale predictions have a compute budget equal to the training

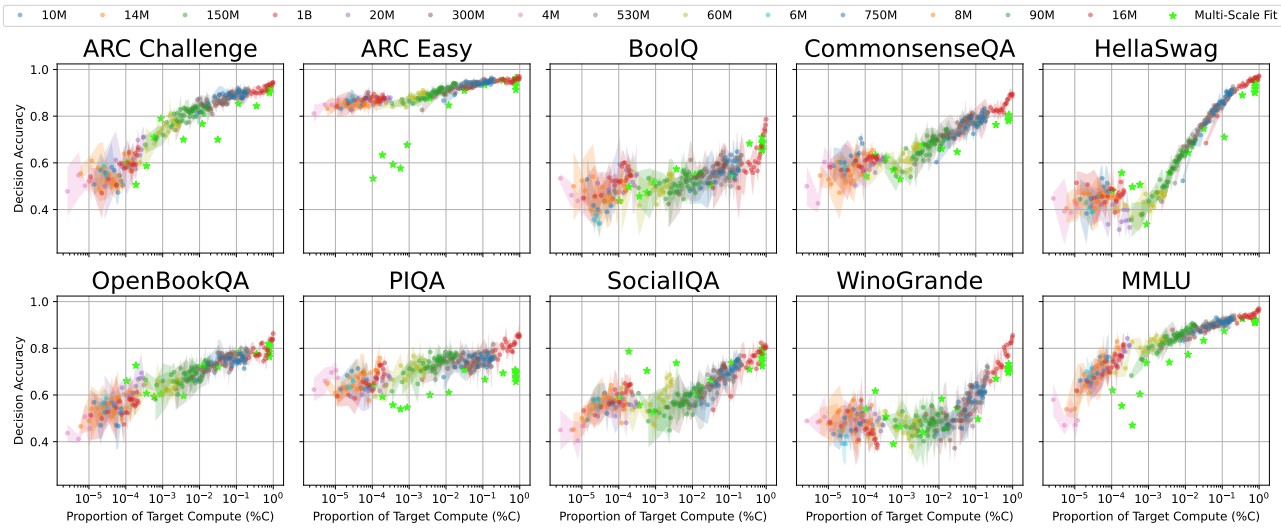

*Figure 2.* Accuracy in pairwise decisions on best data when evaluating on the 10 OLMES tasks with ACCURACY (shown aggregated in Figure 1). Specific tasks have very distinct ranges of sensitivity, with some like ARC Easy being predictable at small scales and others like HellaSwag requiring substantially more compute to predict.

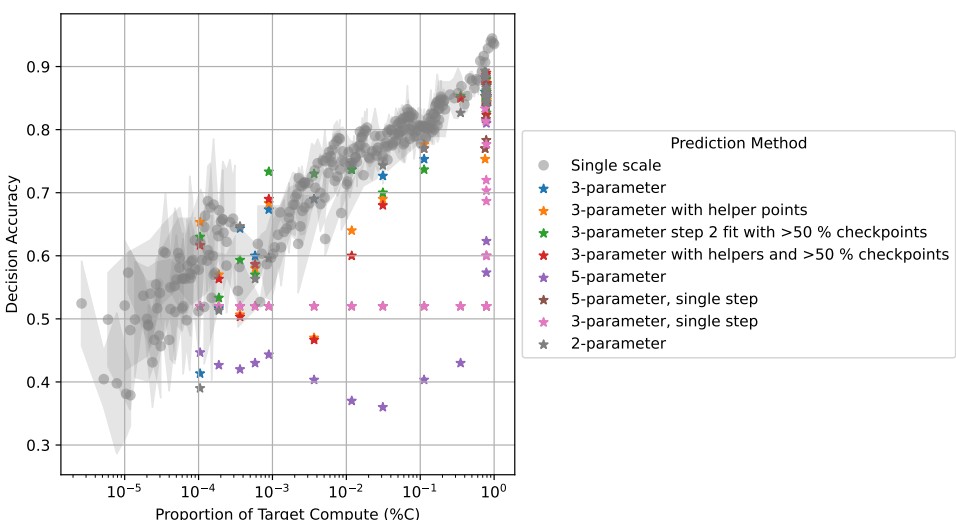

*Figure 3.* Decision accuracy over 8 baseline scaling law variants. At best, these approaches reach only the same compute to decision accuracy frontier as ranking single scale experiments. DATADECIDE can be used to iterate on future scaling law prediction methods.

cost of the model sizes used to make the prediction. We try the following combinations of models sizes: We use $\{\{s_1, \ldots, s_k\} \mid 3 \leq k \leq 14\}$, where $\mathbf{s}$ is the ordered set of sizes, to explore the improvements of progressively adding larger model sizes beyond the minimum 3 required for fitting. We also use $\{\{s_k, \ldots, s_{14}\} \mid 2 \leq k \leq 11\}$ to try removing potentially noisy information from small models. Unlike single scale results, we make only one prediction attempt with the default fully trained random seed, as final checkpoints are required for fitting the first step of these scaling law variants but are not available for all seeds.

Our scaling law approaches vary in the number of parameters fit, using hard coded points to define the minimum and maximum performance, using only the second half of intermediate checkpoints for fitting the second step, or fitting a function directly from compute to accuracy in a single step. Each of the scaling law variants are defined formally in Appendix C. The 2 and 3 parameter variants all achieve among the top decision accuracy.

A priori we know that ranking single scale experiments cannot correctly predict when the scaling trend of one data recipe overtakes another at scales between our small experiments and target scale. Such crossovers bound the decision accuracy of this constant approximation of performance. Nevertheless ranking single scale experiments sets a high baseline decision accuracy, implying relatively little crossover occurs. It is difficult to distinguish evaluation variance from true crossovers, but the scaling trends we empirically observe cross over frequently. Improved future scaling laws may be able to advance the Pareto frontier on DATADECIDE as they are not bound by crossovers.

### 3.3. What proxy metrics give better signal for predictions at small scale?

*At small scales, continuous metrics using the character normalized likelihood of correct or all answer options serve as better or equivalent predictors of decisions than using the same ACCURACY as used at the target scale.*

Figure 4 shows the decision accuracy over different proxy metrics. Here we chose a single length normalization, \*_PER_CHAR. Metrics follow similar trends regardless of length normalization and this one is empirically optimal for most of the tasks that we observe.

Using CORRECT PROB or TOTAL PROB leads to decision accuracy at least as good as any other metric for most small scales. These continuous metrics are simple likelihoods over answer strings. In particular, TOTAL PROB may be interpretable as signal of a model having exposure to the domain of a given task in the form of higher likelihoods on

incorrect but presumably relevant additional answers.

We notice two very distinct types of trends over the different tasks. Either the different proxy metrics are nearly indistinguishable and increase in decision accuracy with compute or CORRECT PROB and TOTAL PROB are flat with respect to scale and the other metrics only rise up to that level of decision accuracy towards the full target compute budget. In the last order of magnitude below the target compute ACCURACY and the other metrics tend to overtake CORRECT PROB and TOTAL PROB, while these two metrics sometimes even decrease in decision accuracy. Notably these other metrics that trend with ACCURACY include continuous metrics that penalize probability assigned to incorrect answers, NORM CORRECT PROB and MARGIN.

### 3.4. How can we make evaluation benchmarks more predictable?

*The decision accuracy on a task is driven in part by low run-to-run variance and a wide spread of performance values for different data recipes. Using CORRECT PROB sees wider spreads or reduced noise for many tasks. Using this metric enables predicting rankings for code tasks that are too hard for accuracy metrics at small scales.*

What underlies differences in decision accuracy when benchmarks and metrics change? The evaluation must separate pairs of data recipes by an amount greater than combined noise from run-to-run variance of each of the pair's runs. In Figure 5, we plot tasks with a given metric using fully trained 150M models over these two characteristics: 1) noise—the standard deviation over 3 random seed runs averaged over all recipes, and 2) spread—the standard deviation among the mean performance of the different data recipes. Each point also shows the decision accuracy. We see that some highly predictable tasks (e.g., MMLU) are characterized by having low run-to-run noise, while others (e.g., ARC Easy) widely spread the different data recipes. We also see that improvements from using CORRECT PROB often align with improvements in one of these two characteristics.

As a practical application of these insights, we demonstrate that a change of proxy metric makes predictable two code tasks (Austin et al., 2021; Chen et al., 2021) that are otherwise too challenging for our small models. Figure 6 shows how decision accuracy goes from trivial to 80% when using CORRECT PROB. The switch of metric allows small models to get above the noise floor for these tasks, while still predicting large-scale accuracy metrics. Notably, two math benchmarks (Lewkowycz et al., 2022; Cobbe et al., 2021) do not see such a benefit. They do however give decision accuracy above 80% if we switch the *target metric* to CORRECT PROB, raising a question for future work to

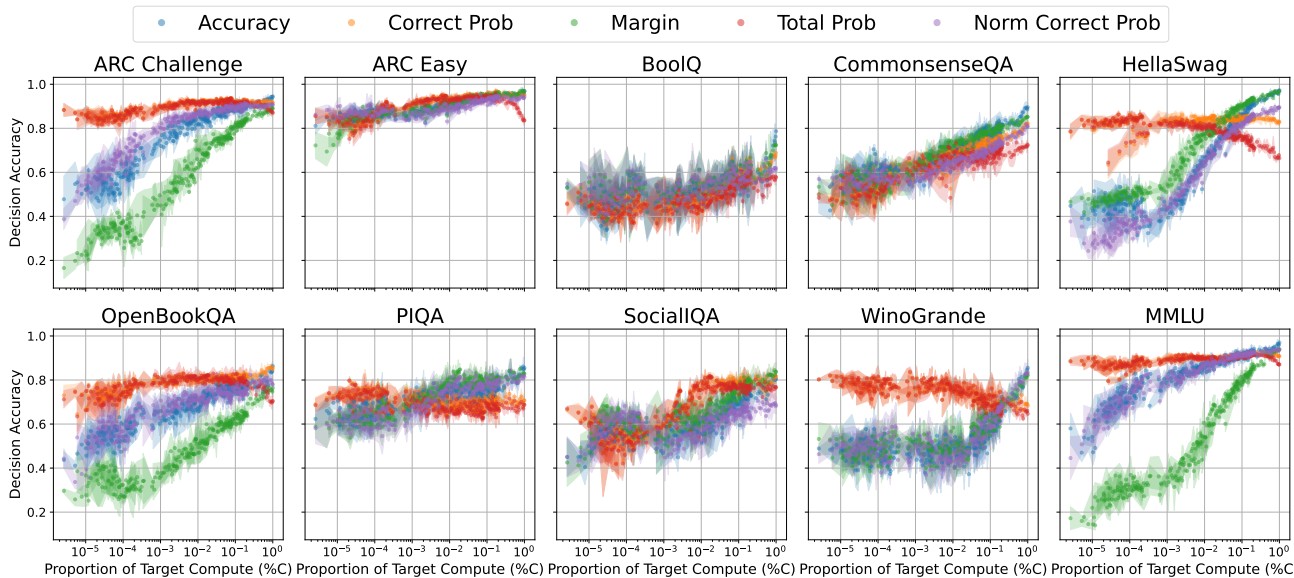

Figure 4. Per-task decision accuracy using character normalized proxy metrics for ACCURACY targets. 5 tasks benefit at smaller scales from using raw likelihood of answers (CORRECT PROB and TOTAL PROB), as opposed to discrete ACCURACY or continuous metrics that penalize probability on incorrect answers (NORM CORRECT PROB, MARGIN).

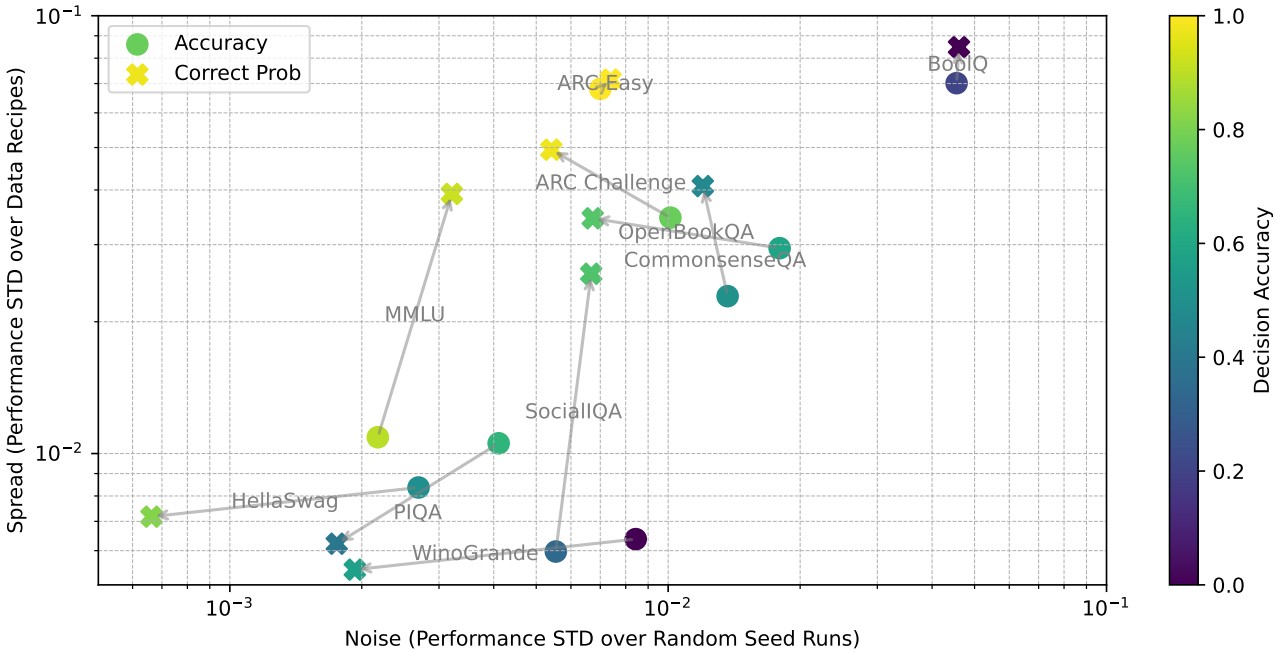

Figure 5. Why do some tasks or metrics get better or worse decision accuracy? At 150M with CORRECT PROB tasks like HellaSwag succeed with low run-to-run variance and tasks like SocialIQA widely spread the performance assigned to different pretraining data.

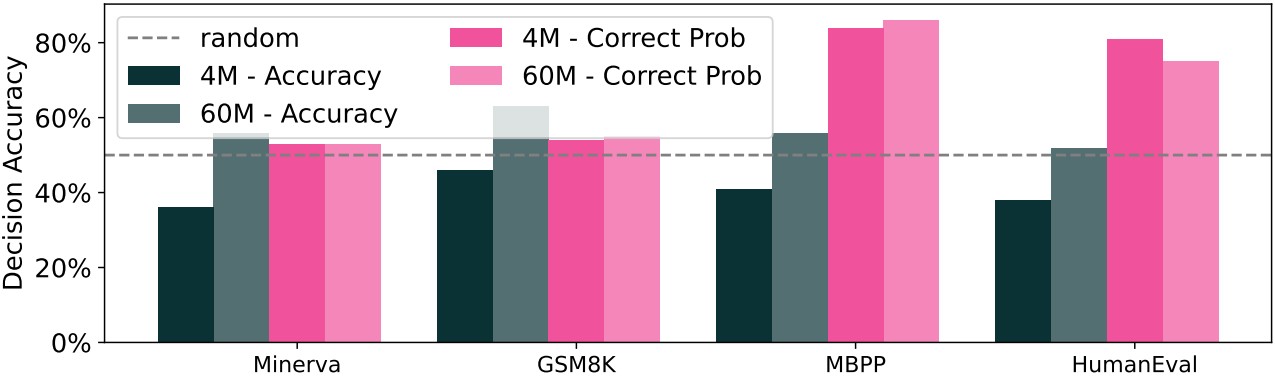

*Figure 6.* Code tasks such as humaneval and MBPP go from trivial decision accuracy to largely predictable when using using continuous CORRECT PROB instead of discrete ACCURACY. Meanwhile common math tasks remain near trivial decision accuracy regardless of metric.

explore whether changing the target metric can be justified.

## 4. Related Work

**Prediction** Much work studies scaling behavior in language models. Initially this focused on predicting LM loss from scale as determined by parameter count and tokens trained (Kaplan et al., 2020; Hoffmann et al., 2022). Special consideration is also given to the case of data constrained scaling (Muennighoff et al., 2023; Goyal et al., 2024). Unlike predicting loss, predicting downstream performance from scale is generally harder (Schaeffer et al., 2024). However, recent work has demonstrated it can be done based on a two step prediction that chains together predictions from scale to loss and loss to downstream performance (Gadre et al., 2024; Bhagia et al., 2024; Dubey et al., 2024), sometimes using training loss (Du et al., 2024) or transferring losses from different data recipes (Brandfonbrener et al., 2024; Ruan et al., 2024). The one line of work targeting pretraining data considers the special case of deciding mixing proportions of several data sources optimized through scaling laws (Kang et al., 2024; Ye et al., 2024). Most relevant to our work, Choshen et al. (2024) consider practical methods for better scaling prediction error such as how much compute to use or whether to include intermediate checkpoints. Orthogonally to these findings, we propose a way to assess the accuracy of decisions made with such predictions.

**Suites over Data Differences** DATADECIDE follows in the footsteps of the Pythia Suite (Biderman et al., 2023) which was the first to offer a controlled comparison of 2 data recipes, using compute scales up to $2 \times 10^{22}$ FLOPs. Subsequent suites have offered 6 data recipes at $9 \times 10^{20}$ scale (Magnusson et al., 2024) and 6 data recipes over a

range of scales up to $10^{21}$ (Brandfonbrener et al., 2024). Our DATADECIDE offers a range of 14 scales up to $7 \times 10^{20}$ FLOPs, while including an order of magnitude more fine-grained data differences. Meanwhile, DCLM also makes extensive use of ranking single scale experiments to drive improvement in data recipes (Li et al., 2024). They release their best data and a model trained on it, but do not release models from their decision making experiments and do not search over multiple recipes at their largest scale. Where their goal is creating a proposed best recipe, our DATADE-CIDE enables the assessment of whether a method for decision making really does find the best among proposed recipes.

## 5. Limitations

The scope of our work is limited to just one ratio of tokens to parameters, 100 or 5× "Chinchilla" optimal ratio (Hoffmann et al., 2022). We believe this captures the typical case, as most models now favor overtraining for inference savings. Due to compute limitations and the need for a standardized set of model configurations over a long period of time in which compute became available for pretraining, we opt for 14 specific configurations from 4M–1B parameter scale. While observations across more configurations would always be better, this must be traded off with exploring the other dimensions of data recipes and random seed reruns. Likewise, while our 25 data recipes is an order of magnitude more than previous suites, there is always the possibility that findings across these will not be representative of future data recipes. In our evaluations we focus on multiple choice tasks with a "cloze" formulation as we find these to be a good fit for our range of scales. Using DATADECIDE, new evaluations can be assessed easily by others without any additional pretraining.

## Acknowledgments

We would like to thank Dave Wadden, Kyle Lo, Valentin Hofmann, and Hannaneh Hajishirzi for fruitful conversations. This material is based upon work supported by the U.S. National Science Foundation under Grant No. 2313998. Any opinions, findings, and conclusions or recommendations expressed in this material are those of the author(s) and do not necessarily reflect the views of the U.S. National Science Foundation. IM is supported by the NSF CSGrad4US Fellowship. PWK is supported by the Singapore National Research Foundation and the National AI Group in the Singapore Ministry of Digital Development and Information under the AI Visiting Professorship Programme (award number AIVP-2024-001) and by the AI2050 program at Schmidt Sciences.

## Impact Statement

Training large language models is computationally expensive, especially when investigating thoroughly over dimensions of pretraining data composition, model scale, random initialization, and data order. The pretraining experiments in our DATADECIDE required approximately 820K H100 GPU hours. We share the benefit of this cost through releasing all of our models, data, and evaluations so that others will not have to repeat this expenditure. Moreover, our findings can guide efficient and cost-effective model development through the application of decision making with small-scale experiments. While DATADECIDE does not present direct ethical concerns beyond opportunity cost, we acknowledge that decisions about pretraining data heavily impact downstream model behavior. We encourage future research to explore potential biases in data selection methods and their implications for models deployed in the real world.

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

## A. Hyperparameters

Table 2 provides OLMo model ladder configurations for all models in DATADECIDE.

## B. Proxy Metric Definitions

Table 3 provides formal definitions for our proxy metrics (§2.5).

## C. Scaling Law Variants

**Baseline 3-parameter fit.** Our default setup (described in §2.2) follows the two-step fit from (Bhagia et al., 2024) and uses Equation 1 to map compute $C$ to task loss $L$, and Equation 2 to map task loss to metric score. This variant fits three parameters $(A, \alpha, E)$ in the first step.

**2-parameter fit.** This is a restricted version of the baseline where the irreducible loss term $E$ is removed from Equation 1, leaving only two parameters:

$$L(C) = \frac{A}{C^\alpha} \quad (4)$$

**5-parameter $(N, D)$ fit.** Instead of modeling loss as a function of compute $C$, this variant uses both number of tokens $N$ and number of parameters $D$ directly in the loss function:

$$L(N, D) = \frac{A}{N^\alpha} + \frac{B}{D^\beta} + E \quad (5)$$

This introduces five parameters: $A$, $\alpha$, $B$, $\beta$, and $E$.

**Single-step prediction.** In this variant, the two-stage fitting procedure is replaced with a single step that directly maps compute $C$ to accuracy:

$$Acc(C) = \frac{a}{1 + \exp\left(-k\left(\frac{A}{C^\alpha} + E - L_0\right)\right)} + b \quad (6)$$

This combines the loss and accuracy mapping into one function.

**5-parameter, single step.** We also test a single-step variant that directly maps from $(N, D)$ to accuracy using a logistic function over the predicted loss. This merges Equations 5 and 2 into:

$$Acc(N, D) = \frac{a}{1 + \exp\left(-\left(\frac{A}{N^\alpha} + \frac{B}{D^\beta} + E\right)\right)} + b \quad (7)$$

This formulation retains the same five parameters from the two-step $(N, D)$ loss function. Following Bhagia et al. (2024), we merge the parameters $k$ and $L_0$ from the second-stage sigmoid into the loss-side parameters $(A, B, E)$, yielding a simplified single-stage fit with 7 total free parameters: $\{A, \alpha, B, \beta, E, a, b\}$.

| Model name | Batch size | Hidden dim. | LR | Model size | Heads | Layers | Training steps | Tokens trained |
|---|---|---|---|---|---|---|---|---|
| 4M | 32 | 64 | 1.4e-02 | 3.7M | 8 | 8 | 5,725 | 0.4B |
| 6M | 32 | 96 | 1.2e-02 | 6.0M | 8 | 8 | 9,182 | 0.6B |
| 8M | 32 | 128 | 1.1e-02 | 8.5M | 8 | 8 | 13,039 | 0.9B |
| 10M | 32 | 144 | 1.0e-02 | 9.9M | 8 | 8 | 15,117 | 1.0B |
| 14M | 32 | 192 | 9.2e-03 | 14.4M | 8 | 8 | 21,953 | 1.4B |
| 16M | 32 | 208 | 8.9e-03 | 16.0M | 8 | 8 | 24,432 | 1.6B |
| 20M | 64 | 192 | 8.4e-03 | 19.1M | 8 | 16 | 14,584 | 1.9B |
| 60M | 96 | 384 | 5.8e-03 | 57.1M | 12 | 16 | 29,042 | 5.7B |
| 90M | 160 | 528 | 4.9e-03 | 97.9M | 12 | 16 | 29,901 | 9.8B |
| 150M | 192 | 768 | 4.2e-03 | 151.9M | 12 | 12 | 38,157 | 15.0B |
| 300M | 320 | 1,024 | 3.3e-03 | 320.0M | 16 | 16 | 45,787 | 30.0B |
| 530M | 448 | 1,344 | 2.8e-03 | 530.1M | 16 | 16 | 57,786 | 53.0B |
| 750M | 576 | 1,536 | 2.5e-03 | 681.3M | 16 | 16 | 63,589 | 75.0B |
| 1B | 704 | 2,048 | 2.1e-03 | 1176.8M | 16 | 16 | 69,369 | 100.0B |

*Table 2.* DATADECIDE uses OLMo's *model ladder* (Groeneveld et al., 2024; OLMo et al., 2025; Bhagia et al., 2024) to programmatically create configurations for 14 model sizes with hyperparameters determined by heuristics in Porian et al. (2024). All models have sequence length of 2024 and MLP ratio of 8. Each configuration is pretrained over 25 data recipes (Table 1). Each recipe and configuration is also trained for 3 random seeds where model sizes < 1B are stopped early at 25% of the compute used to train the 1B model for all but the default seed. Model size is number of non-embedding parameters. Batch size is the number of sequences per batch.

| Metric Name | Equation |
|---|---|
| CORRECT PROB | $\frac{1}{N}\sum_{i=1}^{N} P(c^{(i)}_{\text{correct}} \mid \text{context}_i)$ |
| MARGIN | $\frac{1}{N}\sum_{i=1}^{N} \left( P(c^{(i)}_{\text{correct}} \mid \text{context}_i) - \max_{c' \neq c^{(i)}_{\text{correct}} \in C^{(i)}} P(c' \mid \text{context}_i) \right)$ |
| NORM CORRECT PROB | $\frac{1}{N}\sum_{i=1}^{N} \frac{P(c^{(i)}_{\text{correct}} \mid \text{context}_i)}{\sum_{c \in C^{(i)}} P(c \mid \text{context}_i)}$ |
| TOTAL PROB | $\frac{1}{N}\sum_{i=1}^{N} \sum_{c \in C^{(i)}} P(c \mid \text{context}_i)$ |
| ACCURACY | $\frac{1}{N}\sum_{i=1}^{N} \mathbb{I}\left( \arg\max_{c \in C^{(i)}} P(c \mid \text{context}_i) = c^{(i)}_{\text{correct}} \right)$ |
| *_per_token | $\log(P(c \mid \text{context})) / \text{tokens}(c)$ |
| *_per_char | $\log(P(c \mid \text{context})) / \text{chars}(c)$ |

*Table 3.* Proxy metrics used as alternative inputs to our prediction methods, $C^{(i)}$ is the set of possible continuations for item $i$ and $N$ is the number of items in a benchmark. Each each of the first 5 metrics have *_per_token and *_per_char variants in which likelihoods are normalized as defined in the bottom two rows.

| Scaling Law Variant | Relative Error | Absolute Error |
|---|---|---|
| 3-parameter with helpers and >50% checkpoints | 5.6 | 2.6 |
| 3-parameter with helper points | 6.0 | 2.8 |
| 3-parameter step 2 fit with >50% checkpoints | 5.9 | 2.9 |
| 3-parameter | 6.5 | 3.1 |
| 2-parameter | 6.5 | 3.2 |
| 5-parameter, single step | 42.8 | 17.4 |
| 3-parameter, single step | 42.9 | 42.3 |
| 5-parameter | 230.8 | 65.4 |

*Table 4.* Average prediction error for 1B targets for the different scaling law setups across tasks and recipes on ACCURACY fit to all models but 1B. We see that other than the single step and 5-parameter variants errors are comparable, and these variants also roughly follow the compute-decision frontier in Figure 3.

**Use of helper points.** Following Bhagia et al. (2024), we optionally include an extra point ($L = 0.0$, $Acc = 1.0$) in the second-stage fit. This "helper" point anchors the upper asymptote of the accuracy prediction.

**Filtering early checkpoints.** We experiment with excluding the first 50% of intermediate checkpoints when fitting the second-stage sigmoid. This reduces noise from high-loss early training points and often improves the fit for extrapolation.

**Helpers and $> 50\%$ checkpoints.** Lastly we experiment with combining the previous two techniques on the baseline 3-parameter fit.

**Prediction Error.** We report prediction errors in Table 4 for each setup. As the best scaling laws variants are all roughly comparable to the simple 3-parameter set up, we use this one as our baseline.

