# OpenReview forum: "DataDecide: How to Predict Best Pretraining Data with Small Experiments"
_ICML.cc/2025/Conference — ICML 2025 poster_

### Official Review · Reviewer_B1fY · 2025-03-05

**Overall Recommendation:** 4

**Summary:**

This paper presents DataDos, a suite of experiments to examine the comparison of 25 variously processed corpora across scales by pretraining them with up to 1B models and 100B tokens. It finds that 150M models trained with < 2% compute of 1B targets correctly predict 80% of comparisons; spending compute towards smaller, more completely trained experiments for ranking is cost-effective for overall decision accuracy, but fitting scaling law predictions will provide additional information for extrapolating scales or extreme quality thresholds. Detailed per-task/metric analyses reveal different levels of predicting difficulty across tasks.

**Claims And Evidence:**

Claims about how to predict the best pretraining data across scales are supported by the results of comprehensive experiments.

**Essential References Not Discussed:**

No.

**Experimental Designs Or Analyses:**

The experiment and analysis design is overall sound.

**Methods And Evaluation Criteria:**

The methods and evaluation metrics for pretraining models on different corpora are reasonable and comprehensive.

**Other Comments Or Suggestions:**

I would suggest the authors run experiments on a target scale of 7B because this is likely to be the smallest mainstream LLM scale, but the feasibility definitely depends on the available computational resources the authors have because pretraining is always costly.

The writing also can be polished to make this work more impactful. For example,

- Better arrangement of the order of subsections in Section 3, preferably aligned with Section 2.

- Polishing all Figures to make them more readable.

**Other Strengths And Weaknesses:**

The strengths of this paper lie in its comprehensive experiments and detailed analyses about predicting pretraining data differences over scale.

The main weakness, from my perspective, is the clarity and the organization of paper writing.

**Questions For Authors:**

Mainly about clarity issues.

1. In the discussion of Figure 3, "To identify a generally best metric per task we find the metric which achieves the highest decision accuracy in the most compute budgets." What do you mean by that? I suppose the decision accuracy is only measured in the target budget. Also, "average decision accuracy for a metric over its best outcomes", what do you mean by "best outcomes"?

2. You mention "150 extrapolated comparisons" in 3.3. Why 150? I cannot interpret this number from Figure 5.

**Relation To Broader Scientific Literature:**

Scaling law has shown effective promises in guiding the optimal usage of computation. This paper examines the data comparison law across scales, which can further provide guidance and insights into the future development of pretraining data curation, especially extrapolating beyond small-scale experiments.

**Theoretical Claims:**

This paper does not make theoretical claims and focuses mostly on the empirical part.

---

> ### Author Rebuttal · Authors · 2025-04-01
>
> We thank the reviewer for their detailed clarifying questions and are happy they found our work has “comprehensive experiments and detailed analyses.”
>
> **Expanded results**
>
> We appreciate the reviewer’s encouragement to polish our paper. Since submission we have fully revised our paper and extended our suite with more small runs and examined additional benchmarks. Updated figures, which we hope are more readable, can be viewed [here](https://anonymous.4open.science/api/repo/tmp-pdf-4938/file/ICML_responce-2.pdf).
> |  | Initial Submission | Current Revision |
> |---|---|---|
> | Total Models | 375 models | 1,050 models |
> | Model Scales | 5 (from 150M to 1B) | 14 (from 4M to 1B) |
> | Random Seeds | 3 (<1B: up to 10% of target compute; 1B: 100%) | 3 (<1B: up to 25% of target compute; 1B: 100%) |
> | Evaluation Benchmarks | OLMES | OLMES + Code (MBPP, HumanEval) + Math (GSM8K, Minerva) |
> | Scaling Law Methods | 1 variant with 3 windows of model sizes used for compute | 8 variants, with 23 windows of model sizes down to 4 orders of magnitude less than target compute |
>
> We find that **the conclusions to our research questions hold over these new results**: 1) Compute trades off with decision accuracy and different tasks get better signal for decisions. 2) 8 scaling law methods fail to outperform single-scale ranking; future work can use DataDos to iterate on these. 3) Continuous metrics over raw likelihood are always better at small compute; differences due to which length normalization is used are smaller. 4) Low run-to-run variance and large spread between recipes explain success; we can also use proxy metrics to get beyond noise floor for tasks like code benchmarks.
>
> **Clarity and Organization**
>
>
> Following the reviewer’s suggestion, we will move the discussion of scaling laws ahead of proxy metrics in section 3 (results) to better align with the order of topics in section 2 (methods). Also regarding the reviewer’s question 1: In the third figure in our submission we identified best metrics based on which achieves the highest decision accuracy over all of the small compute sizes used for prediction. Indeed, the *target* budget remains the same (1B 5xC) but we consider which metric gets best decision accuracy for the most small *experimental* budgets. Regarding the “best outcomes,” this quantity would be better defined as measuring the average decision accuracy over tasks and scales when using the best metric for each task. In our revised figure (#4), we instead use a new, more direct figure to visualize the relationship of compute to decision accuracy for each type of metric with a specific length normalization. Regarding question 2: 150 is the number of combinations visualized in the former Figure 5, since the number of rows is halved to save space. Thanks to the reviewer’s feedback we’ll be sure to make this more clear.

---

### Official Review · Reviewer_M26A · 2025-03-06

**Overall Recommendation:** 4

**Summary:**

The paper "Data Differences over Scale (DataDos) Suite: How to Predict Best Pretraining Data with Small Experiments" presents an empirical study on the predictability of pretraining data decisions at large scales using small-scale experiments. The authors conduct controlled pretraining experiments across 25 corpora, training models up to 1B parameters and 100B tokens, and introduce the DataDos Suite, which provides open-source models, data, and evaluations. The study finds that single-scale experiments, rather than multi-scale extrapolations, are the most efficient for predicting the best pretraining data, achieving 80% accuracy with only 2% of the compute budget of the target scale. The research also highlights the predictability of certain benchmarks, such as MMLU and ARC, using small-scale experiments while others, like CSQA, require more compute. The paper positions DataDos as a resource for researchers to explore data selection and scaling trends.

**Claims And Evidence:**

Main claims:

- C1: Single scale experiments are more efficient for performance prediction than multiple scales. Figure 2 clearly shows that single ranking provides better decision accuracy than multiple scale extrapolation at various compute budget and tasks. I consider the evidence for the claim is sufficient.

- C2: No single metric gives better signal consistently. I consider the evidence (Sec 3.2) is sufficient.

**Essential References Not Discussed:**

Unknown.

**Experimental Designs Or Analyses:**

1. The experimental design appears to be robust to me, covering 25 data recipes, 5 model scales, and 3 random seed variations to control for initialization effects (target model only).

2. Models are trained using fixed architectures while varying datasets and compute budgets to isolate scaling relation and ensure a fair comparison.

3. Insightful findings and analysis includes
- Mixed results for proxy metrics choice;
- limitation of single ranking method in the case of predicting performance orders of magnitude.
- Mixed results for benchmark signals.

I think the paper will benefits from more discussion on the roots of mixed results. For examples, what drives the discrepancy of benchmarks as a signal for large model performance? Why some benchmarks are saturated while others are not? Does it mean single rank method predict well simply because of the saturation of the benchmark (too easy?)?

**Methods And Evaluation Criteria:**

**Benchmarks**

OLMES suite of 10 multi-choice QA benchmarks is used. It contains common and standard benchmarks for LLM evaluation.

**Metrics**

The study uses both performance prediction errors and decision accuracy to reflect the practical consequence of pretraining data prediction.

**Methods**
The study effectively isolates the impact of scaling decisions by holding model architecture constant while varying training data and compute. It presents large scale experiments consists of a suite of 375 models using 25 data recipes, 5 model scales, and 3 random seeds for initialization and data order. I also find using 1B as the target model is sufficient for generalization purpose given the compute budget.

**Other Comments Or Suggestions:**

None

**Other Strengths And Weaknesses:**

**Strengths**:
- Open-source contribution of models, datasets, and evaluations.

**Weakness**
- Limited depth analysis on the cause of scaling law failure and single rank success.
- nitpick: lack of sensitivity analysis for hyperparameter choices.

**Questions For Authors:**

- When performance is not reported per task, do you report the average of OLMES?
- Does the conclusion hold when using the benchmark as a whole, by group? This tests the sensitivity of the conclusion to benchmark choices.

**Relation To Broader Scientific Literature:**

The study builds on prior work in language model scaling laws and data selection, referencing key papers such as Kaplan et al. (2020) and Hoffmann et al. (2022). It differentiates itself by focusing specifically on data selection rather than general model scaling, bridging a gap in the literature.

**Theoretical Claims:**

None

---

> ### Author Rebuttal · Authors · 2025-04-01
>
> We thank the reviewer for their clear and helpful feedback and supporting that “using 1B as the target model is sufficient for generalization purpose given the compute budget.”
>
> **Expanded results**
>
> Since our submission we have enriched our analysis and extended our suite with more small runs and examined additional benchmarks. Updated figures can be viewed [here](https://anonymous.4open.science/api/repo/tmp-pdf-4938/file/ICML_responce-2.pdf).
> |  | Initial Submission | Current Revision |
> |---|---|---|
> | Total Models | 375 models | 1,050 models |
> | Model Scales | 5 (from 150M to 1B) | 14 (from 4M to 1B) |
> | Random Seeds | 3 (<1B: up to 10% of target compute; 1B: 100%) | 3 (<1B: up to 25% of target compute; 1B: 100%) |
> | Evaluation Benchmarks | OLMES | OLMES + Code (MBPP, HumanEval) + Math (GSM8K, Minerva) |
> | Scaling Law Methods | 1 variant with 3 windows of model sizes used for compute | 8 variants, with 23 windows of model sizes down to 4 orders of magnitude less than target compute |
>
> We find that **the conclusions to our research questions hold over these new results**: 1) Compute trades off with decision accuracy and different tasks get better signal for decisions. 2) 8 scaling law methods fail to outperform single-scale ranking; future work can use DataDos to iterate on these. 3) Continuous metrics over raw likelihood are always better at small compute; differences due to which length normalization is used are smaller. 4) Low run-to-run variance and large spread between recipes explain success; we can also use proxy metrics to get beyond noise floor for tasks like code benchmarks.
>
> **Explanations of better decisions**
>
> We appreciate the reviewer’s encouragement to provide further discussion of the “roots” of differing decision accuracy. In our revised figure (#5) we provide an analysis of how decision accuracy of different tasks and metrics can be explained in part by low run-to-run variance and a wide spread of performance values for different data. Using the correct probability proxy metric sees wider spreads for many tasks, though some instead see reduced noise. The reviewer’s questions about saturation in particular are interesting, as this could either lead to easy decisions (if some recipes saturate at lower performance than others) or to noisy decisions (if all recipes saturate around a similar value by the target scale). However, in our suite we do not observe saturation on any of the benchmarks we examine. Instead we see near trivial performance on Boolq, which leads to it having low decision accuracy as its target rankings are mostly determined by noise. Thus we do not recommend it for use on predictions over the scales we examine.
>
> Likewise we appreciate the reviewer’s encouragement to provide further analysis of scaling laws vs. single ranking. Our smaller results let us consider a wider range of compute budgets for scaling law fits, and we elevate our 8 additional scaling law variants (previous appendix line 770) to the body of our paper (see new figure #3). Our finding that these baseline scaling laws do not outperform single scale ranking holds over the breadth of sizes and variants. The latest results show that the single ranking approach is strong at even smaller compute budgets than we previously found, and we believe future work is required to ascertain the specific ways in which scaling laws fail despite getting low prediction error.
>
> **Aggregate benchmark analysis**
> Regarding the reviewer’s question: when figures do not name a specific task for downstream performance, the quantity is the macro average of OLMES (line 202). We find that our claims hold when aggregating: intermediate checkpoints decide as well as compute equivalent final models, single scale experiments outperform scaling laws, continuous metrics outperform discrete ones at small scales. As we examine in new (and former) figure (#2) slope and range of the positive relationship between compute and decision accuracy depends a lot on which benchmark is used, but the other claims we make hold.

---

### Official Review · Reviewer_fgMX · 2025-03-14

**Overall Recommendation:** 2

**Summary:**

The paper introduces the DATADOS Suite, an extensive experimental framework designed to guide pretraining data decisions for large language models using small-scale experiments. By systematically exploring 25 data recipes (varying in sources, deduplication, filtering, and mixing) across 5 model scales with a fixed token-to-parameter ratio, and repeating experiments with 3 different random seeds (yielding 375 runs in total), the authors aim to predict which data recipes will yield the best downstream performance when scaled up to a 1B parameter, 100B token regime. It further introduces a set of continuous proxy metrics (e.g., normalized correct probability, margin) that improve decision accuracy over standard discrete metrics, reporting that even small models (150M parameters) trained with less than 2% of the target compute can correctly predict 80% of pairwise winners among data recipes. I believe the paper is well-motivated because the full scaling law for data quality decision can be very computational expensive and the benchmark score doesn't make sense if the model is only trained on small amount of data and doesn't emerge.

**Claims And Evidence:**

I think the main claim and contribution of this paper is that small‐scale pretraining experiments (e.g., using 150M–parameter models) can reliably predict which data recipes will perform best when scaled up to target levels (1B parameters, 100B tokens). They report an 80% decision accuracy in predicting the winner between data recipe pairs.

The paper conducted extensive experiments covering 25 recipes and multiple scales, with detailed reporting of decision accuracy and proxy metric performance. However, while the reported 80% decision accuracy is promising, the reliance on a fixed token-to-parameter ratio and a narrow range of model sizes (150M to 1B) may limit the robustness of these conclusions when applied to other settings.

**Essential References Not Discussed:**

N/A

**Experimental Designs Or Analyses:**

The paper covers 25 data recipes across a fixed range of model sizes and including multiple seeds offers an extensive dataset for analysis. The clear comparison between ranking and scaling law approaches provides actionable insights. However, the experimental design is constrained to a single token-to-parameter ratio and a limited range of scales, which might not capture the full spectrum of scaling behavior (for example, we compare two dataset, one dataset is of high quality but has limited tokens, another is of low quality but has enough tokens, in this case we have to predict the performance in a larger data scale). Moreover, the evaluation is restricted to OLMES multiple choice tasks, raising questions about generalizability to other evaluation paradigms or domains. One key concern is that we believe the results are valuable as a summary but for new tasks/data, people can be unsure about how reliable the analysis results are.

**Methods And Evaluation Criteria:**

The DATADOS is proposed to estimate the dataset performance/quality using relatively small model and data while keeping high accuracy. They also provide the proxy metrics for different primary benchmarks. The paper is well motivated and the results are very helpful as a guidance or technique report. However, the methodology is still holistic - one key issue is that if we have new benchmark (e.g. shopping_mmlu) and a new dataset/recipe (e.g. nemotron-cc), we have to re-run all the experiments to get the estimation but the estimation is not well-explained and may be difficult to extrapolate (compared to scaling law). For the proxy metrics, I agree the continuous metrics can be much better indicator for small scale experiments before the model emerges. However, there is a lack of analysis on how these proxy metrics are robust across different data recipes and models - if we use another models/data recipe, will the proxy keep accurate. Otherwise, some theoretical bound/guarantee can also be helpful.

**Other Comments Or Suggestions:**

Please kindly refer to the previous sections.

**Other Strengths And Weaknesses:**

The experiments are extensive and the analysis is detailed. However, besides the empirical study, it would be great if the paper can further elaborate on 1) the reliance/theoretical support of the methods; 2) how well the methods/findings can be transferred to different model families and tasks; 3) if it is difficult to guarantee the former properties, it is great to make the proposed methods a easy to use tool.

**Questions For Authors:**

Please kindly refer to the previous sections.

**Relation To Broader Scientific Literature:**

The work is well-situated within the literature on scaling laws, data curation, and pretraining data selection, building on foundational studies  as well as recent empirical efforts.

**Theoretical Claims:**

The authors note that only 4.7% of scaling trends “crossover” at the current scales, but warn that as compute increases, more crossovers may occur.  It would be great if the author can provide more theoretical analysis/insights for the observations. It remains vulnerable to noise in small-scale experiments. A deeper discussion of how sensitive the extrapolations are to such noise and how uncertainty in the scaling law fits is handled would be valuable.

---

> ### Author Rebuttal · Authors · 2025-04-01
>
> We thank the reviewer for their thoughtful questions and are glad they found our “paper is well motivated and the results are very helpful.”
>
> The reviewer writes that “the main contribution of this paper is that small-scale pretraining experiments (e.g., using 150M–parameter models) can reliably predict which data recipes will perform best.” We highlight that our paper has two main contributions: 1) We release an extensible framework, easily updated with new prediction methods, benchmarks, and small models. 2) We provide practical recommendations based on empirical study of decision accuracy that we observe to be reliable over wide ranges of settings, such as model sizes (now from 4M to 1B parameters) and benchmarks (including knowledge QA, commonsense, and reasoning, as well as math and code now).
>
> **Expanded results**
>
> We have extended our suite with more small runs and evaluations. Updated figures can be viewed [here](https://anonymous.4open.science/api/repo/tmp-pdf-4938/file/ICML_responce-2.pdf).
> |  | Initial Submission | Current Revision |
> |---|---|---|
> | Total Models | 375 models | 1,050 models |
> | Model Scales | 5 (from 150M to 1B) | 14 (from 4M to 1B) |
> | Random Seeds | 3 (<1B: up to 10% of target compute; 1B: 100%) | 3 (<1B: up to 25% of target compute; 1B: 100%) |
> | Evaluation Benchmarks | OLMES | OLMES + Code (MBPP, HumanEval) + Math (GSM8K, Minerva) |
> | Scaling Law Methods | 1 variant with 3 windows of model sizes used for compute | 8 variants, with 23 windows of model sizes down to 4 orders of magnitude less than target compute |
>
> We find that **the conclusions to our research questions hold over these new results**: 1) Compute trades off with decision accuracy and different tasks get better signal for decisions. 2) 8 scaling law methods fail to outperform single-scale ranking; future work can use DataDos to iterate on these. 3) Continuous metrics over raw likelihood are always better at small compute; differences due to which length normalization is used are smaller. 4) Low run-to-run variance and large spread between recipes explain success; we can also use proxy metrics to get beyond noise floor for tasks like code benchmarks.
>
> **Ease of use and tooling**
>
> The reviewer encourages us to make DataDos easy to use, for instance adding a “new benchmark (e.g. shopping_mmlu).” In our public release of 1K models, 25 corpora, and code for pretraining, evaluation, prediction, and decision accuracy (line 104) we will include step by step documentation for adding new tasks, prediction methods, and small pretraining experiments. We will host all checkpoints as native models on Hugging Face. Running new tasks can be as simple as adding a new argument to the open source evaluation framework that we leverage. Our aim is that researchers with a range of compute can build on our suite: 1) by trying new prediction methods with lightweight manipulations of our results CSV such as smoothing or curve fitting, 2) adding new benchmarks over our released checkpoints for just the cost of inference, 3) or even pretraining additional small models for a fraction of the target model cost (e.g., the 9 additional model sizes we trained since submission cost only ~2% of our total compute).
>
> **Reliable and generalizable claims**
> The reviewer wrote that our experiments covered “a narrow range of models (150M to 1B)”, so we’ve significantly expanded our results to include 9 additional even smaller model sizes as small as  4M, or ~0.3% of the 1B compute. We continue to find consistent trends of compute to decision accuracy that support our claims for our 4 research questions (line 50). We agree that evaluation noise is a concern at small model scale, so our work is the first in this area to train multiple random seeds, even for the largest model sizes (line 152) We use the mean of these to make our decision targets more reliable. The reviewer asked about how are claims generalize, so we added four new heldout benchmarks and found that our preferred continuous proxy metric also leads to better decision accuracy when target performance is above the noise floor (new Figure #6).
>
> **Focus on exploring data difference**
>
> Our study investigates dimensions of differences not yet studied extensively; we explore 25 different data recipes, an order of magnitude more than previous suites.   We chose to spend our compute budget by overtraining (5x the Chinchilla multiplier) many models (14 different sizes) and evaluating many checkpoints per model to estimate what performance would be like with different token-parameter ratios. Our conclusions are very consistent across intermediate and final checkpoints, suggesting that different ratios might give similar results, but we leave that to future work to confirm.

---

### Decision · Program_Chairs · 2025-05-01

**Decision:**

Accept (poster)

**Comment:**

This paper presents a well-executed empirical study showing that small-scale pretraining experiments can reliably predict the best data recipes for large-scale LLM training. All reviewers acknowledge the value of the extensive experiments and open-source contributions. While one reviewer raises concerns about generalization, the authors addressed these with expanded experiments and clearer analyses. I recommend acceptance of the paper.